# SAFE EXPLORATION IN DOSE FINDING CLINICAL TRIALS WITH HETEROGENEOUS PARTICIPANTS

## ABSTRACT

In drug development, early phase dose-finding clinical trials are carried out to identify an optimal dose to administer to patients in larger confirmatory clinical trials. Standard trial procedures do not optimize for participant benefit and do not consider participant heterogeneity, despite consequences to the health of participants and downstream impacts to under-represented population subgroups. Additionally, many newly investigated drugs do not obey modelling assumptions made in common dose-finding procedures. We present Safe Allocation for Exploration of Treatments (SAFE-T), a procedure for adaptive dose-finding that works well with small samples sizes and improves the utility for heterogeneous participants while adhering to safety constraints for treatment arm allocation. SAFE-T flexibly learns models for drug toxicity and efficacy without requiring strong prior assumptions and provides final recommendations for optimal dose by participant subgroup. We provide a preliminary evaluation of SAFE-T on a comprehensive set of realistic synthetic dose-finding scenarios, illustrating the improved performance of SAFE-T with respect to safety, utility, and dose recommendation accuracy across heterogeneous participants against a comparable baseline method.

## 1 INTRODUCTION

New drugs and treatments are generally first investigated in early phase dose-finding studies, which aim to assess safety and provide recommended dose levels for future study. Dose-finding trials often include a small number of participants due to safety concerns and difficulties with participant recruitment. Increasingly, researchers are promoting adaptive trial methods (in contrast to rule-based methods), where trial parameters may change based on ongoing participant outcomes, which can improve both the efficiency of a trial and the outcomes experienced by trial participants (Villar et al., 2015; Riviere et al., 2018).

Dose-finding studies commonly use the rule-based 3+3 method, which has been criticized for its inefficiency (Kurzrock et al., 2021), or the adaptive continual reassessment method (CRM), which requires a pre-selected parametric model and strict prior assumptions (Wheeler et al., 2019). These methods assume that the patient population is homogeneous and do not account for possible variations in drug toxicity and efficacy due to patient heterogeneity. Due to these assumptions, alongside persisting inequalities in subject selection for clinical trials (Steinberg et al., 2021; Chien et al., 2022), optimal drug doses derived from such trials are often not generalizable to women, who remain under-represented in early-phase trials (Özdemir et al., 2022). Multiple analyses have shown that women experience a far greater risk of adverse drug effects across all drug classes as compared to men (Zucker & Prendergast, 2020; Unger et al., 2022). Recent work has found that race and ethnicity can also impact drug response and has highlighted the inadequate understanding of drug safety and efficacy across under-studied racial and ethnic populations (Ramamoorthy et al., 2021; Dickmann & Schutzman, 2018). In addition to the issue of participant heterogeneity, newly developed drugs may not adhere to standard prior assumptions, particularly that dose-toxicity and dose-efficacy levels are monotonically increasing. Innovative therapies may follow plateauing or unimodal efficacy functions (Wages et al., 2018; Zhang et al., 2006).

While complex factors surrounding inequality in clinical trials must be addressed to improve outcomes for under-represented populations, we make a small contribution with our adaptive dose-finding procedure, Safe Allocation for Exploration of Treatments (SAFE-T). SAFE-T models toxi-

city and efficacy dose-response curves with multi-output Gaussian processes that capture variations between participant subgroups and are flexible to different dose-response shapes. Using these dose-response estimates, SAFE-T balances safety and exploration in allocation of doses to trial participants, resulting in both improved participant utility and final dose recommendation accuracy.

**Related work.** Recent works in machine learning for dose-finding trials have mostly concentrated on multi-armed bandit methods (Aziz et al., 2020), with some including explicit safety constraints (Lee et al., 2020; Shen et al., 2020; Wang et al., 2021). These works consider dose-finding scenarios with monotonically increasing toxicity and efficacy curves, with Aziz et al. (2020); Shen et al. (2020) also providing separate algorithms for plateauing efficacy curves. Lee et al. (2020) propose methods that address heterogeneous populations with pre-defined subgroups. We note that Lee et al. (2020) addresses a dose-finding setting most similar to ours; however, their method includes an un-safe burn-in period and uses a parametric model that is not flexible to different toxicity curve shapes. The problem of safe exploration has been addressed more generally in the context of bandits (Kazerouni et al., 2016) and Bayesian optimization with Gaussian processes (Sui et al., 2015; 2018). The issue of fair exploration (with respect to population subgroups) for bandit algorithms has also been investigated in Raghavan et al. (2018); Baek & Farias (2021).

## 2 PROBLEM STATEMENT

### 2.1 DOSE-FINDING SETUP

We examine early phase dose-finding trials where $N$ trial participants are sequentially, at timestep $t$, allocated one of $K$ discrete doses, indexed by $k \in \mathbb{K} = \{1, ..., K\}$, with dose levels $d_k \in \mathbb{D}$ representing the dosage values. Participants belong to a known subgroup $s \in \mathbb{S} = \{1, ..., S\}$, arriving at rates $\pi_s$. For doses $d$, $g_s(d)$ defines true toxicity probabilities and $f_s(d)$ defines true efficacy probabilities by subgroup $s$. At each timestep, a participant of subgroup $h_t \in \mathbb{S}$ is allocated a dose of index $m_t \in \mathbb{K}$ based on a specified *selection rule*. Following dose allocation, we observe the binary toxicity outcome $Y_t \sim Ber(g_{h_t}(d_{m_t}))$, with $Y_t = 1$ indicating an adverse reaction; and the binary efficacy outcome $X_t \sim Ber(f_{h_t}(d_{m_t}))$, with $X_t = 1$ indicating effective treatment.

Trials using a model-based methodology will aim to accurately estimate the dose-toxicity response relationship, $\hat{g}_s(d)$, which describes the probability of a toxic event, and sometimes the dose-efficacy response relationship, $\hat{f}_s(d)$, which describes the probability of effective treatment, at doses $d \in \mathbb{D}$ for subgroups $s \in \mathbb{S}$. Adverse side effects are categorized as toxic events and patient responses are categorized as effective treatment based on previous clinical knowledge and will typically be formalized prior to implementation of a clinical trial. Trial practitioners specify a target toxicity threshold (TTL), $\tau_T$, which is the highest acceptable probability of a toxic event. Similarly, $\tau_E$ is also specified as the lowest acceptable probability of efficacy for a dose. A dose $d$ for a patient in subgroup $s$ is thus in an acceptable safe range when $g_s(d) \leq \tau_T$ and $f_s(d) \geq \tau_E$.

At the conclusion of a trial, a *recommendation rule* is used to select a final *dose recommendation* $\hat{d}_{s,N}$, which should be equivalent to the *optimal dose* $d_s^*$ for each subgroup $s$. These doses would be examined in larger, downstream clinical trials that focus on determining the efficacy of drugs. Commonly used dose-finding methods select the maximum tolerated dose as the optimal dose, defined as $\hat{d}_{s,N} = \arg\max_{d_k:\hat{g}_s(d) \leq \tau_T} \hat{g}_s(d)$. However, this recommendation rule would fail in cases where the dose-efficacy curve plateaus or is unimodal. SAFE-T provides an alternative method for selecting optimal doses that works well to determine optimal doses across differing curve shapes.

### 2.2 PROBLEM CONSTRAINTS

We discuss the realistic constraints and objectives of an algorithm that can be used for dose-finding trials. Dose-finding trials may incorporate **heterogeneous patients** belonging to differing pre-defined subgroups. Each subgroup may adhere to transformed dose-toxicity and dose-efficacy curves, although it may not be previously known what the difference may be (Thomas et al., 2018). As such, we desire an algorithm that can flexibly learn differences between in the dose-response relationships of patient subgroups. Our algorithm should provide **accurate final dose recommendations** across subgroups, where $\hat{d}_{s,N} = d_s^*, \forall s \in \mathbb{S}$. The **safety** of trial participants is of paramount concern. We aim to minimize safety constraint violations and also avoid using a burn-in period as

done in some multi-armed bandit solutions for dose-finding (Lee et al., 2020; Shen et al., 2020), where the algorithm initializes by selecting each dose in succession regardless of safety. While standard dose-finding procedures do not consider **efficacy** for trial participants, we incorporate it into our methods in order to improve participant outcomes. Recent work in ML has argued that in healthcare settings, we should aim to **maximize expected utility** rather than institute fairness constraints or objectives that may reduce utility (Pfohl et al., 2022). As such, we do not provide explicit fairness constraints or objectives in our methodology, but assess our performance with respect to utility and safety across participant subgroups. The majority of early phase trials include a **small sample size**, often below 50 participants (Can, 2022; Huang et al., 2015). An effective algorithm must be able to learn dose-response relationships for toxicity and efficacy efficiently, with few samples.

## 3 THE SAFE-T ALGORITHM

---

**Algorithm 1** SAFE-T Algorithm

---

1: **Input:** patient subgroups $\mathbb{S}$, dose indices $\mathbb{K}$, number of patients $N$, safe expansion timesteps $N_0$, toxicity threshold $\tau_T$, efficacy threshold $\tau_E$, GP prior for multi-output toxicity function $\hat{g}(d)$, GP prior for multi-output efficacy function $\hat{f}(d)$, utility function $U(p_e, p_t)$,
2: **Initialize:** $t \leftarrow 1, \mathbb{B}_{s,0} \leftarrow \emptyset \forall s \in \mathbb{S}$
3: **while** $t \leq N$ **do**
4: $\quad b = \begin{cases} 0, & \text{if } \mathbb{B}_{h_t, t-1} = \emptyset \\ \max(\mathbb{B}_{h_t, t-1}), & \text{otherwise} \end{cases}$
5: $\quad \mathbb{A}_{h_t, t} \leftarrow \mathbb{B}_{h_t, t-1} \bigcup \{b+1\}$
6: $\quad$ **if** $t < N_0$ **then**
7: $\quad\quad \mathbb{M}_{h_t, t} \leftarrow \{k \in \mathbb{A}_{h_t, t} \,|\, \mu_{\hat{g}_{h_t}(d_k)} \leq \tau_t\}$
8: $\quad\quad$ **if** $\max(\mathbb{M}_{h_t, t}) \notin \mathbb{B}_{h_t, t-1}$ **then**
9: $\quad\quad\quad m_t \leftarrow \max(\mathbb{M}_{h_t, t})$
10: $\quad\quad$ **else**
11: $\quad\quad\quad m_t \leftarrow \arg\max_{k \in \mathbb{M}_{h_t, t}} c_t(d_k)$
12: $\quad\quad$ **end if**
13: $\quad$ **else**
14: $\quad\quad \mathbb{M}_{h_t, t} \leftarrow \{k \in \mathbb{A}_{h_t, t} \,|\, u_t(d_k) \leq \tau_t\}$
15: $\quad\quad m_t \leftarrow \arg\max_{d_k \in \mathbb{M}_{h_t, t}} EI(d_k)$
16: $\quad$ **end if**
17: $\quad$ Observe outcomes $X_t, Y_t$
18: $\quad$ Update $\hat{g}(d), \hat{f}(d)$
19: $\quad \mathbb{B}_t \leftarrow \mathbb{B}_{t-1} \bigcup \{m_t\}$
20: $\quad t \leftarrow t + 1$
21: **end while**
22: $\mathbb{M}_{s,N} \leftarrow \{k \in \mathbb{B}_{s,N} \,|\, u_t(d_k) \leq \tau_t\} \forall s \in \mathbb{S}$
23: $\hat{k}_{s,N} \leftarrow \arg\max_{k \in \mathbb{M}_{s,N}} U(\hat{f}_s(d_k), \hat{g}_s(d_k)) \forall s \in \mathbb{S}$
24: **Output:** $\hat{d}_{s,N} \leftarrow d_{\hat{k}_{s,N}} \forall s \in \mathbb{S}$

---

**Overview.** We present pseudocode for SAFE-T in Algorithm 1 (Appendix **??**). SAFE-T provides *selection rules* for dose allocation during the course of a clinical trial and a final *recommendation rule* at trial completion for selection of optimal doses by subgroup for future study. During the trial, allocation occurs in two stages (with respective selection rules) to better balance safety and learning: a safe exploration stage, where allocation to unexplored doses is encouraged; and a Bayesian optimization phase, where allocation is safely optimized with respect to efficacy. SAFE-T runs for a pre-specified $N$ participants, with the first stage occurring over a pre-specified $N_0$ timesteps, where $N_0 < N$. Initial participants (for each subgroup) are first assigned to the lowest dose, $d_1$.

SAFE-T models the dose-response functions of toxicity, $\hat{g}_s(d)$, and efficacy, $\hat{f}_s(d)$ as two separate multi-output Gaussian processes (GP), using the linear model of co-regionalization (Journel & Huijbregts, 1976), with the correlated outputs of each respective Gaussian process corresponding

to each subgroup $s$. The GPs are trained using stochastic variational inference (Hensman et al., 2014). While SAFE-T requires priors set on certain hyperparameters, the use of Gaussian processes eliminates the need for specifying the explicit parametric form of the dose-responses, as is done in the commonly used continual reassessment method and other proposed model-based dose-finding methods (Wheeler et al., 2019; Lee et al., 2020; Shen et al., 2020). GPs allow for flexibility of the shape of the dose-response relationships, which is beneficial as efficacy may take on a monotonically increasing, plateauing, or unimodal shape. We are also able to take advantage of GP confidence intervals for dose allocation and informative safety constraints.

At the conclusion of a trial, SAFE-T provides a *recommendation rule* for determining the optimal dose for each subgroup to be used in future study. This rule uses a notion of utility concerning dose-finding trials as proposed by Thall & Cook (2004) and further used in (Koopmeiners & Modiano, 2014). It was proposed for use in dose allocation throughout a trial; however, we use it for both the final dose recommendation rule and post-hoc performance analysis. It is a weighted $L_p$ norm that evaluates the trade-off between toxicity and efficacy. We use the slightly modified variation proposed by Koopmeiners & Modiano (2014): $U(p_E, p_T) = 1 - \left( \left( \frac{p_T}{\tau_T} \right)^p + \left( \frac{1-p_E}{1-\tau_E} \right)^p \right)^{\frac{1}{p}}$; $p_E$ refers to probability of efficacy and $p_T$ refers to probability of toxicity. The parameter $p$ is set by an elicitation procedure from clinical trial practitioners (Appendix A.1).

**During trial: Safe exploration stage.** During the safe exploration stage, SAFE-T maintains a set of doses that have been previously sampled, $\mathbb{B}_{s,t}$, for each subgroup $s$. An available set of doses, $\mathbb{A}_{s,t} = \mathbb{B}_{s,t} \bigcup \{\max(\mathbb{B}_{s,t}) + 1\}$, includes all doses that have been previously sampled and the next highest dose (unless the highest dose has been sampled already). $\mathbb{B}_{s,t}$ is empty at the beginning of the trial; a patient belonging to a subgroup that has not yet been encountered is assigned to the lowest dose, $d_1$. The safe set of doses, $\mathbb{M}_{s,t} = \{k \in \mathbb{A}_{s,t} \,|\, \mu_{\hat{g}_s(d_k)} \leq \tau_t\}$, for each subgroup $s$, is determined with respect to the mean of the GP posterior on toxicity for doses in the available set. SAFE-T allocates the highest safe dose available, if this dose has not been previously sampled by subgroup $s$. If all safe doses have been previously sampled, SAFE-T selects the dose with the largest confidence interval on $\hat{g}_s(d)$. Confidence interval widths are referenced by $c_t(d_k) = 2\beta\sigma_{\hat{g}_s(d_k)}$ for each dose $d_k$, where $\beta$ is a scalar hyperparameter. Thus, dose $m_t$ is selected as follows:

$$m_t = \begin{cases} \max(\mathbb{M}_{h_t,t}), & \text{if } \max(\mathbb{M}_{h_t,t}) \notin \mathbb{B}_{h_t,t} \\ \arg\max_{k \in \mathbb{M}_{h_t,t}} c_t(d_k), & \text{otherwise} \end{cases}.$$

**During trial: Safe optimization stage.** During the rest of the trial, SAFE-T selects the dose from the safe set of doses with the highest expected improvement on efficacy. First, we define the upper bound on the confidence interval for toxicity as $u_t(d_k) = \mu_{\hat{g}_s(d_k)} + \beta\sigma_{\hat{g}_s(d_k)}$. While the available set of doses, $\mathbb{A}_{s,t}$, is defined as in the previous stage, the safe dose set, $\mathbb{M}_{s,t} = \{k \in \mathbb{A}_{s,t} \,|\, u_t(d_k) \leq \tau_t\}$, is defined slightly differently in this stage, now incorporating the confidence intervals on the GP posterior on toxicity. SAFE-T then allocates the dose $m_t = \arg\max_{k \in \mathbb{M}_{h_t,t}} EI(d_k)$, $EI(x)$ referring to expected improvement, a commonly used acquisition function (Jones et al., 1998).

**After trial: Final dose recommendation.** At the conclusion of the trial, SAFE-T selects a final recommended dose for each subgroup based on a utility measure, $U(p_e, p_t)$, that incorporates the final posteriors on the GP toxicity and efficacy functions. The safe dose set is composed of all doses that have been allocated during the trial with upper confidence bounds on toxicity below the toxicity threshold: $\mathbb{M}_{s,N} = \{k \in \mathbb{B}_{s,N} \,|\, u_N(d_k) \leq \tau_t\}$. The final recommended dose for each subgroup is the dose with the maximum utility out of the safe dose set: $\hat{k}_{s,N} = \arg\max_{k \in \mathbb{M}_{s,N}} U(\hat{f}_s(d_k), \hat{g}_s(d_k))$; $\hat{d}_{s,N} = d_{\hat{k}_{s,N}}$.

## 4 PRELIMINARY RESULTS

We compare SAFE-T to the C3T algorithm proposed by Lee et al. (2020), as it the only related work that also addresses heterogeneous patients in dose-finding trials. We assess performance across 18 synthetic dose-finding scenarios that comprehensively evaluate performance with respect to differing toxicity (monotonically increasing and plateauing) and efficacy (monotonically increasing, plateauing, unimodal) curve shapes and realistic variations between subgroups, as well as possible edge cases. We work with a population of $N = 51$, $N_0 = 18$, with 2 subgroups arriving at $\boldsymbol{\pi} = [0.5, 0.5]$. The reported metrics are averages over 100 trials of each method and averaged across the trial par-

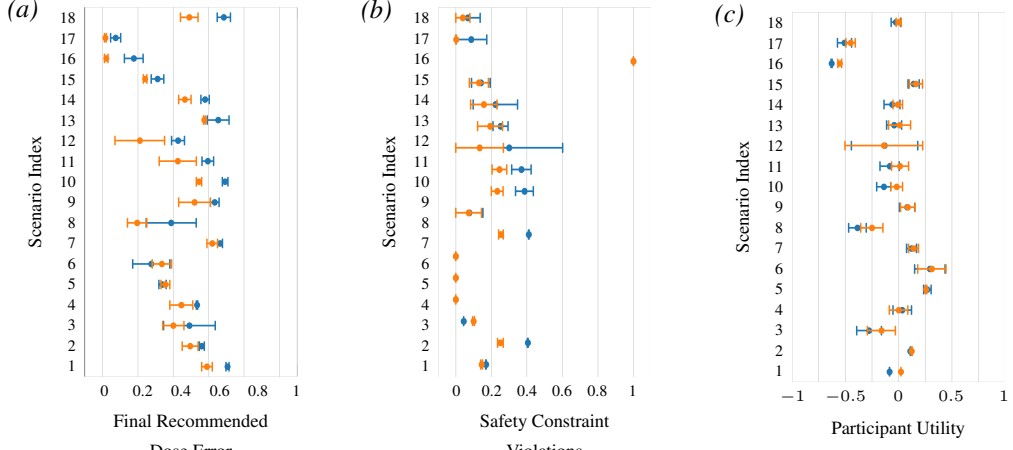

Figure 1: **SAFE-T (orange) consistently outperforms C3T (blue)** with *(a)* lower or similar dose error in all scenarios, *(b)* fewer or similar number of safety constraint violations in 17/18 scenarios, and *(c)* higher or similar participant utility in all scenarios. Circular points represent the metric mean across the 2 subgroups, while line caps represent the metric for each subgroup. Further experiment details are shown in Appendix A.2

ticipants. Results are shown in Figure 1; SAFE-T appears in orange while C3T appears in blue. SAFE-T consistently outperforms C3T in all three categories: final recommended dose error, which assesses whether the recommended dose is equivalent to the true optimal dose; safety constraint violations, which records the number of times a dose with a true toxicity probability greater than the toxicity threshold is allocated to a patient; and participant utility, which is assessed post-hoc based on the dose allocated to each participant. We note that while the utility measure is used to determine final dose recommendations in SAFE-T, it is **not** used during the dose allocation procedure; it is thus a notable result that SAFE-T maintains high utility across many scenarios.

## 5 CONCLUSION

In this paper, we present a method for conducting safe dose-finding trials while maintaining high recommended dose accuracy and participant utility. Our algorithm is constructed to be compatible with the realistic constraints of a dose-finding trial, including effectiveness with small sample sizes and heterogeneous participants. Currently, SAFE-T is limited to settings with binary outcome variables and also adheres to common simplifying assumptions, such as that outcomes are observed without delay. Future work could consider more complex scenarios, such as continuous outcomes, delayed outcomes, multiple outcomes, and missing data (as patient dropout can be common in clinical trials).

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

## A  APPENDIX

### A.1  DEFINITION OF UTILITY

In SAFE-T, we adopt the notion of utility concerning dose-finding trials as proposed by Thall & Cook (2004) and further used in (Koopmeiners & Modiano, 2014). It was proposed for use in dose allocation throughout the trial; however, we use it for the final dose recommendation rule and post-hoc performance analysis. *Thall utility* allows for intuitive interpretation of the toxicity and efficacy trade-off and also lends well to practitioner input. It is a weighted $L_p$ norm that evaluates the trade-off between toxicity and efficacy. We use the slightly modified variation proposed by Koopmeiners & Modiano (2014):

$$U(p_E, p_T) = 1 - \left( \left( \frac{p_T}{\tau_T} \right)^p + \left( \frac{1 - p_E}{1 - \tau_E} \right)^p \right)^{\frac{1}{p}} \tag{1}$$

with $p_E$ referring to probability of efficacy and $p_T$ referring to probability of toxicity, while $\tau_T$ is the maximum toxicity threshold and $\tau_E$ is the minimum efficacy threshold.

The parameter $p$ is determined by setting the utility to 0 and plugging in a midpoint, $(p_E^*, p_T^*)$, elicited from practitioners, that defines the curvature of the contour:

$$1 = \left( \left( \frac{p_T^*}{\tau_T} \right)^p + \left( \frac{1 - p_E^*}{1 - \tau_E} \right)^p \right)^{\frac{1}{p}} \tag{2}$$

## A.2 Experiment Details

In our experiments, we assess the performance of SAFE-T as compared to the C3T algorithm proposed in (Lee et al., 2020). A notable aspect of C3T is that it allows for patient skipping if the patient budget, $B$, is less than the maximum number of timesteps, $N$. We do not include this aspect in our comparison (by simply setting $B = N$), as the difficulties of recruiting for dose-finding trials and ethical considerations in subject selection indicate that skipping patients would be unlikely in a realistic setting (Brøgger-Mikkelsen et al., 2020). In addition, C3T requires a pre-specified dose skeleton of expected toxicity probabilities. A dose skeleton, as described in Wheeler et al. (2019), is used to determine dose labels that will fit the domain of the parametric toxicity model well. While the experiments in (Lee et al., 2020) use the underlying true toxicity probabilities of their synthetic dose-finding scenarios to determine a dose skeleton, we include random noise in our dose skeleton priors, as prior toxicity estimates are unlikely to be perfectly accurate in practice.

To mimic a realistic setting (Wheeler et al., 2019), for our experiments of SAFE-T patients arrive in cohorts of size 3 and models are updated following outcomes observed from all cohort patients. However, the use of cohorts is not discussed in (Lee et al., 2020) so participants are allocated doses one at a time. In addition, the unsafe burn-in period required by C3T is kept in the experiments although this procedure would likely not be possible in a practical setting.

SAFE-T models dose-toxicity and dose-efficacy with multi-output Gaussian process, using the linear model of co-regionalization (LMC) (Journel & Huijbregts, 1976). The LMC model assumes that each output dimension is a linear combination of $Q$ learned latent functions, $\boldsymbol{g}(\cdot) = [g^{(1)}(\cdot), \ldots, g^{(Q)}(\cdot)]$:

$$\boldsymbol{f}_s(\boldsymbol{x}) = \sum_{i=1}^{Q} a^{(i)} g^{(i)}(\boldsymbol{x}) \tag{3}$$

In our setting, each output dimension (also referred to as a *task*) corresponds to a subgroup. Thus, our Gaussian processes learn subgroup representations that are composed of underlying latent functions, with $s \in \{1, \ldots, S\}$, $S = 2$, and $Q = 3$.

Due to the small sample sizes we expect to work with, we set many of the hyperparameters of the Gaussian processes ahead of time. These hyperparameters have been manually tuned to work with the standard value ranges of dose-finding trials. Hyperparameters remain the same across all 18 test scenarios for comparability. Both the toxicity and efficacy GPs use constant mean functions (we set mean$= -0.3$ for toxicity and mean$=-0.1$ for efficacy) and the radial basis function kernel (RBF kernel) as the covariance function (we set lengthscale$= 4$ for toxicity and lengthscale$=2$ for efficacy). We also set the matrix $\boldsymbol{A}$, with $Q$ rows and $S$ columns, which is composed of the coefficients $a_s^{(i)}$ of the LMC model to $\left( \begin{smallmatrix} 1.0 & 0 \\ 0.2 & 0.2 \\ 0 & 1.0 \end{smallmatrix} \right)$. These hyperparameters were manually tuned and the high performance maintained across all 18 distinct synthetic scenarios suggests that they are applicable across variable dose-finding settings. However, they could be further tuned for even further improved performance for specific settings. For example, the lengthscale parameter can be informed by the range of dose values, which will be known ahead of a trial.

The true underlying toxicity and efficacy probabilities for each of the 18 scenarios are shown in the figures below. Green shows the probabilities for subgroup 0 and pink shows the probabilities for subgroup 1. If only a pink curve is shown, that means both subgroups have the same true probabilities. The red dot shows the optimal dose for subgroup 0 and the blue dot shows the optimal dose for subgroup 0. If no optimal dot is seen, it means that there is no optimal dose (either all doses or too toxic or not effective enough). The horizontal black lines show respectively the maximum toxicity threshold and minimum efficacy threshold.

*(1a)* 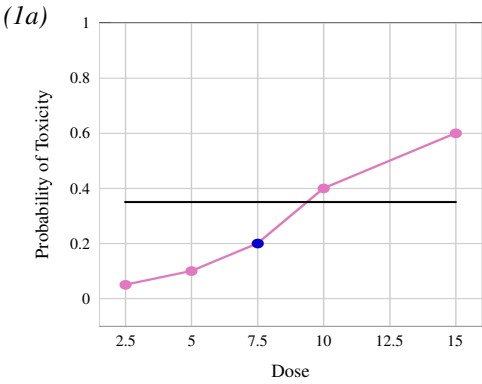

*(1b)* 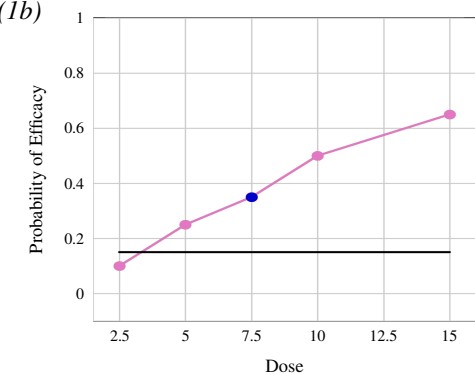

*(2a)* 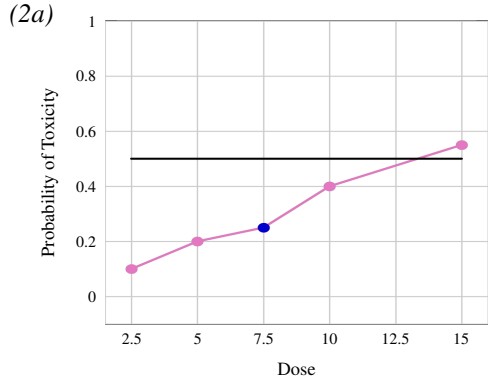

*(2b)* 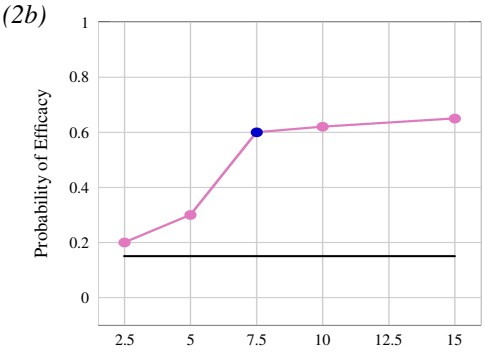

*(3a)* 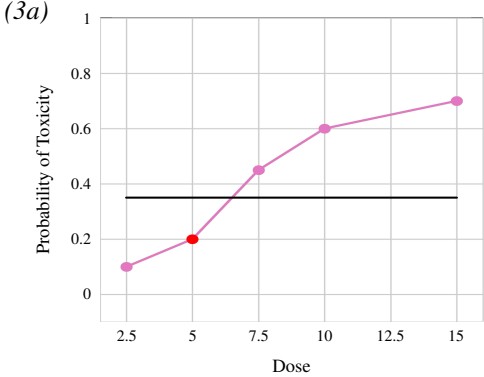

*(3b)* 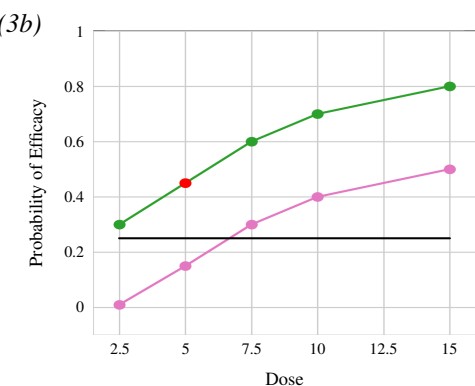

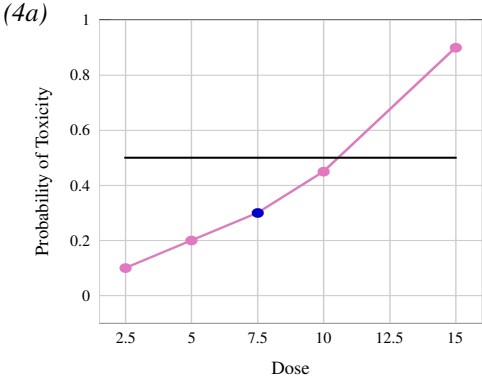

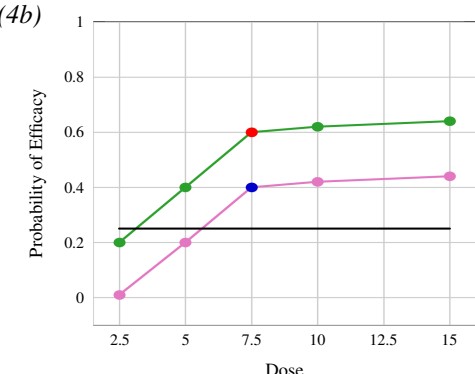

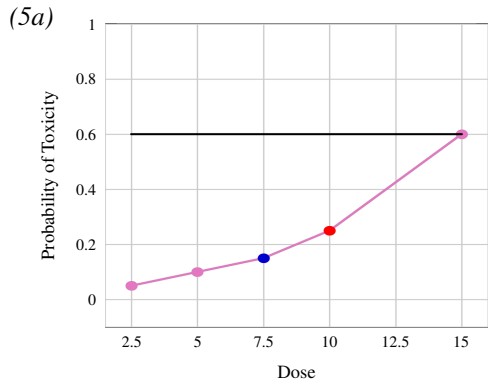

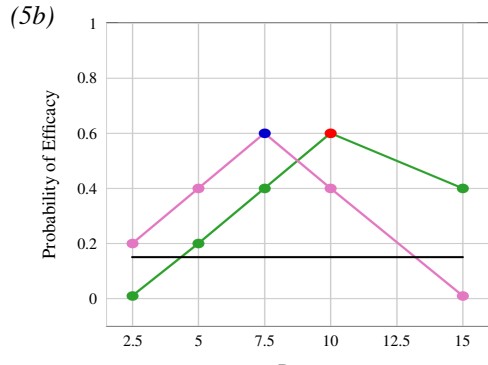

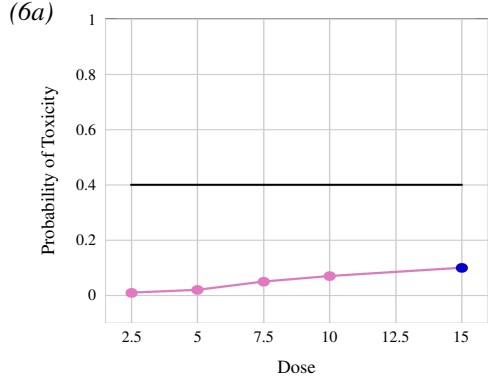

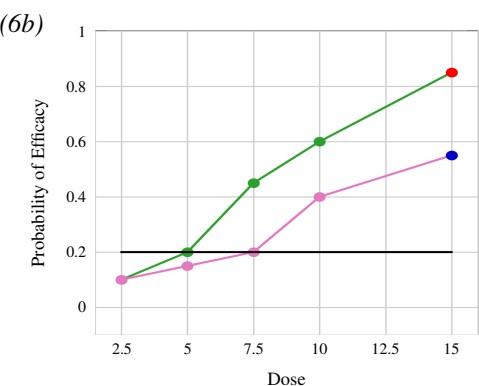

*(7a)*

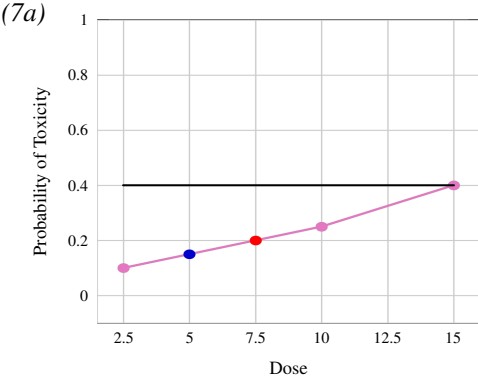

*(7b)*

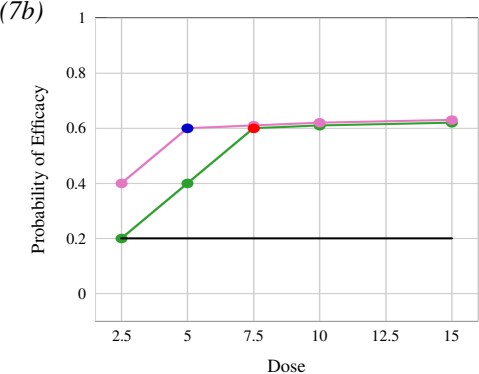

*(8a)*

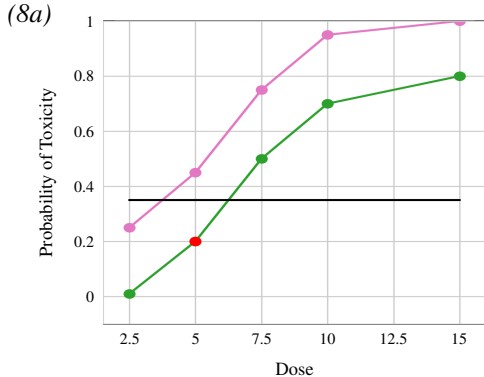

*(8b)*

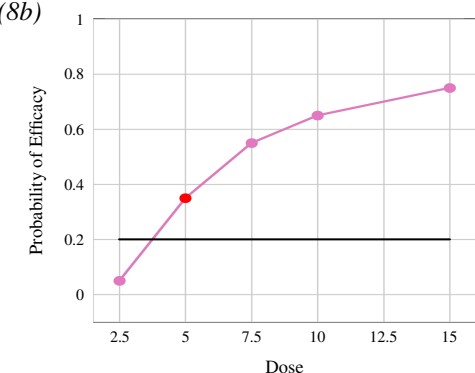

*(9a)*

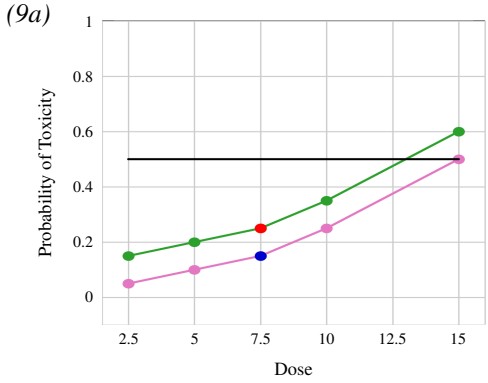

*(9b)*

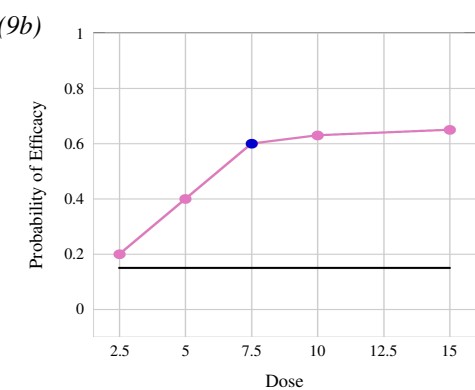

*(10a)* 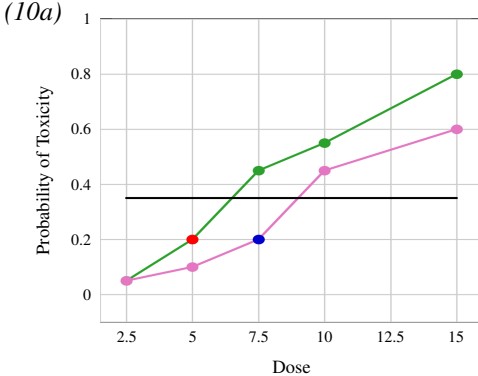

*(10b)* 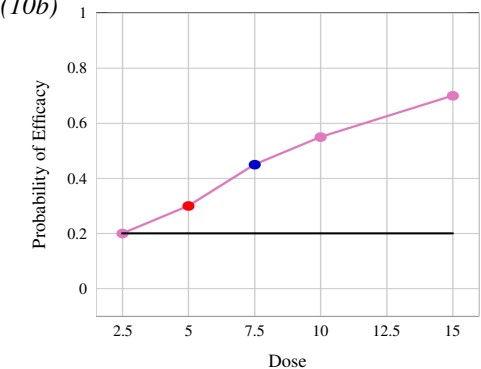

*(11a)* 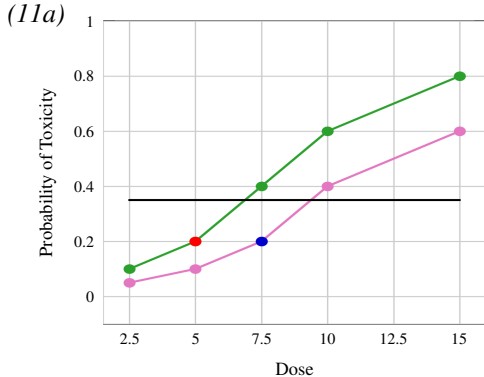

*(11b)* 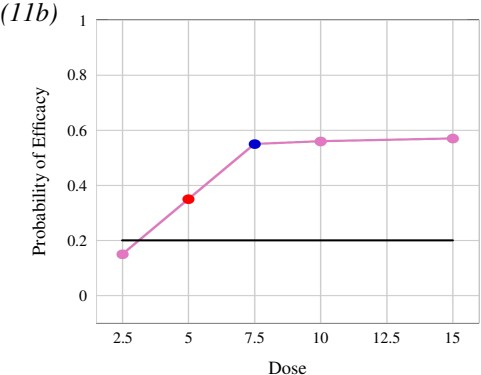

*(12a)* 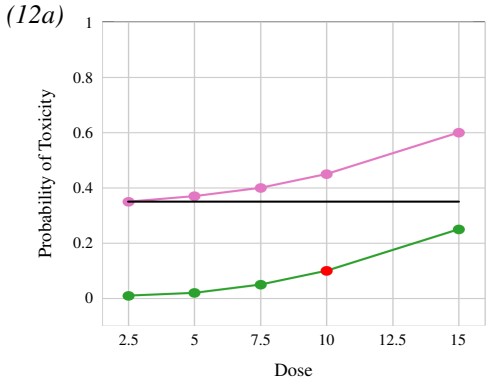

*(12b)* 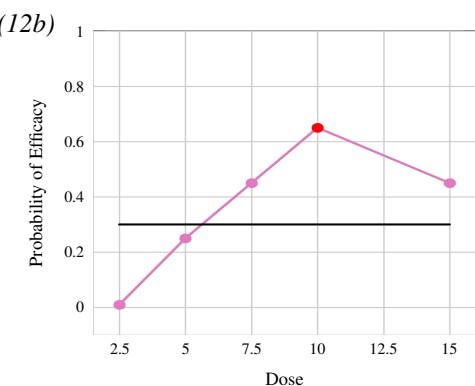

*(13a)* 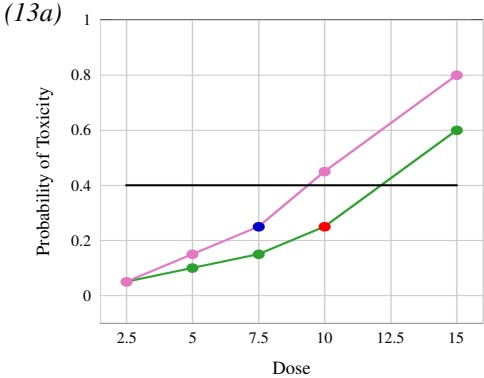

*(13b)* 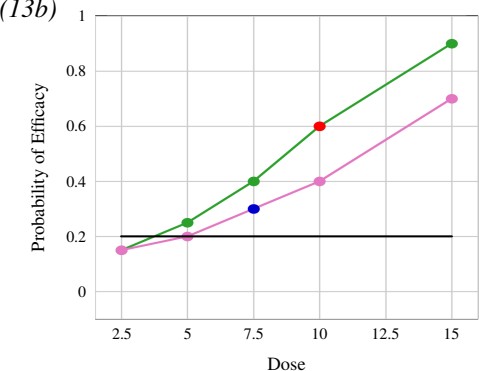

*(14a)* 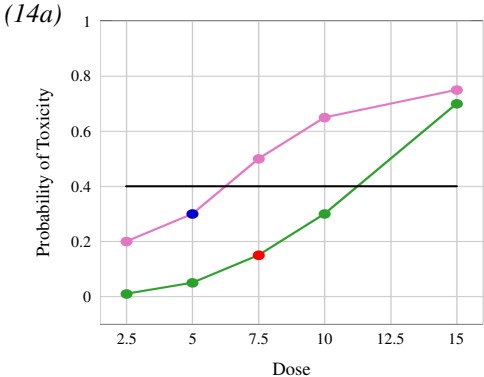

*(14b)* 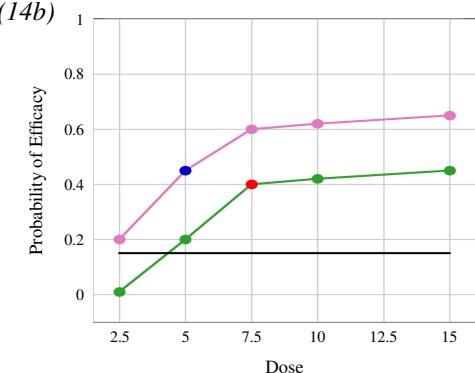

*(15a)* 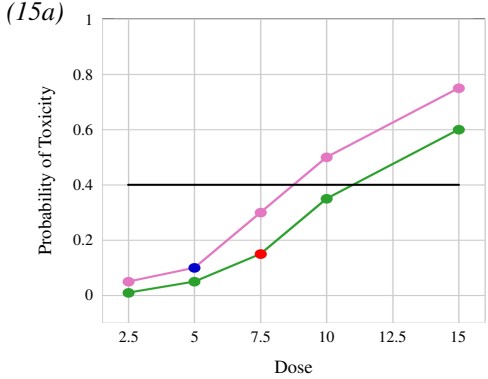

*(15b)* 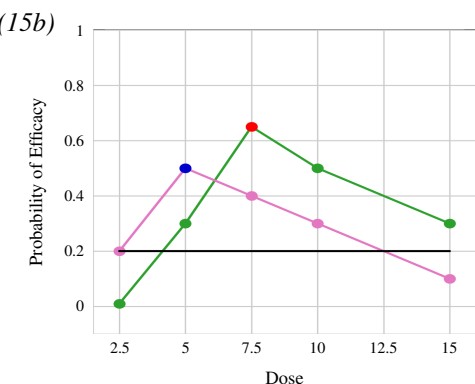

*(16a)*
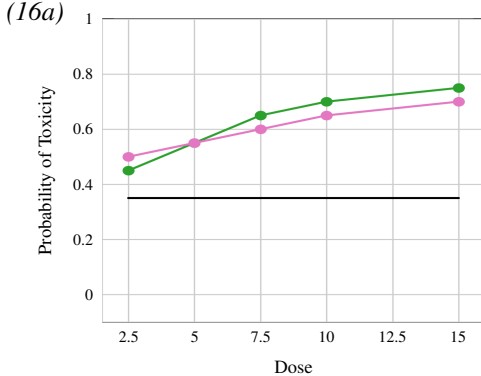

*(16b)*
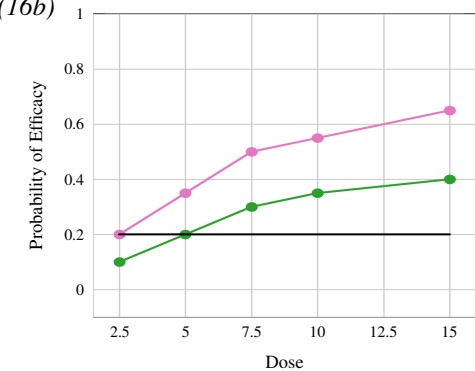

*(17a)*
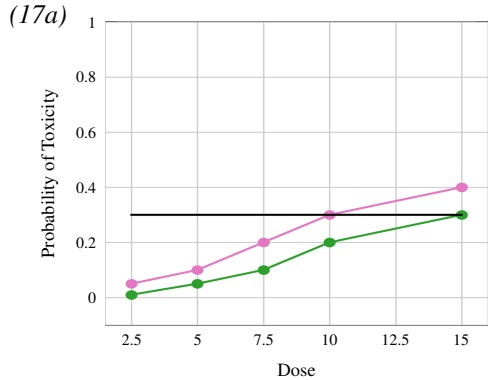

*(17b)*
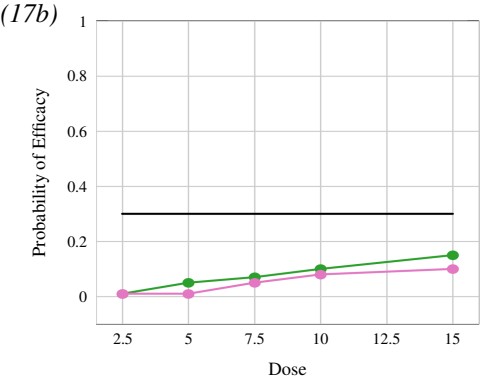

*(18a)*
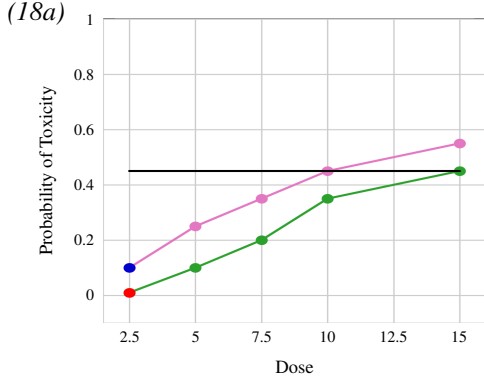

*(18b)*
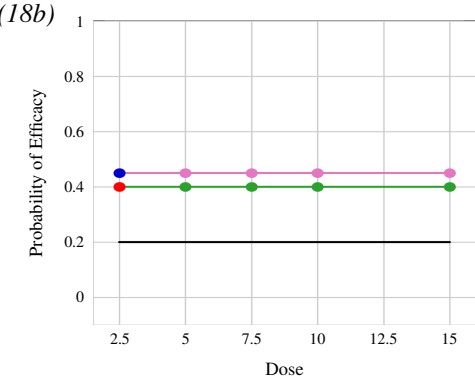

