# OpenReview forum: "Safe Exploration in Dose Finding Clinical Trials with Heterogeneous Participants"
_ICLR.cc/2023/Workshop/TML4H — ICLR 2023 Workshop TML4H Oral_

### Official Review · Reviewer_xceg · 2023-03-01
**An interesting work that bring machine learning to clinical trial design**

**Rating:** 10
**Confidence:** 3

**Review:**

This paper describes a novel adaptive algorithm for optimizing dose rules in clinical trials that work with small sample sizes and considers the utility for heterogeneous participants. This work tackles a novel application of machine learning that was uncommon in previous ML for healthcare works. This paper would therefore be of sufficient interest to the community.

Strengths:
- The topic of this work is novel while it may bring both scientific and social benefits for the stakeholders of clinical trials.
- The background and the methodology are described in sufficient detail.
- The experiment results are presented in sufficient detail.

Weakness:
- The real-world challenges (technical & administrative) for applying algorithm-based dose distribution rules are expected to be further discussed.

---

### Meta-Review · Area_Chair_h4To · 2023-03-06

**Recommendation:** Accept (Poster)
**Confidence:** 4

**Metareview:**

The manuscript presents a new method for adaptive dose-finding that works well with small sample sizes and improves the utility for heterogeneous participants. It is evaluated on a comprehensive set of realistic synthetic dose-finding scenarios, outperforming the baseline method. This work can contribute to the community of AI drug discovery.
1)	The method is full of novelty and will be interesting to researchers in the field of AI drug discovery.
2)	The manuscript is well organized, with a good introduction of the background, a detailed literature review of related works, and good description of the proposed method.

Cons:
1)	The content of the manuscript seems too short (Just 4 pages without reference). Please move some important parts in a brief form from the Appendix to the manuscript.
2)	Please add Conclusion section in the manuscript and briefly discuss the limitations of the proposed method and future directions.